# Assessing T-Cell Immunity in Kidney Transplant Recipients with Absent Antibody Production after a 3rd Dose of the mRNA-1273 Vaccine

**DOI:** 10.3390/ijms232012333

**Published:** 2022-10-15

**Authors:** Maria Infantino, Aris Tsalouchos, Edda Russo, Selene Laudicina, Valentina Grossi, Barbara Lari, Maurizio Benucci, Lorenzo Stacchini, Amedeo Amedei, Patrizia Casprini, Danilo Villalta, Pietro Claudio Dattolo, Mariangela Manfredi

**Affiliations:** 1Immunology and Allergology Laboratory Unit, S. Giovanni di Dio Hospital, 50143 Florence, Italy; 2Nephrology and Dialysis Unit Firenze II, Santa Maria Annunziata Hospital, 50139 Florence, Italy; 3Department of Experimental and Clinical Medicine, University of Florence, 50134 Florence, Italy; 4Rheumatology Unit, S. Giovanni di Dio Hospital, 50143 Florence, Italy; 5Department of Health Science, University of Florence, 50134 Florence, Italy; 6Clinical Pathology Laboratory, S. Giovanni di Dio Hospital, 50143 Florence, Italy; 7Immunology and Allergology Laboratory Unit, S-Maria degli Angeli Hospital, 33170 Pordenone, Italy

**Keywords:** interferon-gamma release assay, COVID-19, kidney transplantation, T cells

## Abstract

The vulnerable population of kidney transplant recipients (KTRs) are low responders to COVID-19 vaccines, so specific immune surveillance is needed. The interferon-gamma (IFN-γ) release assay (IGRA) is effective in assessing T cell-mediated immunity. We assessed SARS-CoV-2-directed T cell responses in KTRs with absent antibody production after a third dose of the mRNA-1273 vaccine, using two different IGRAs. A cohort of 57 KTRs, who were actively followed up, received a third dose of the mRNA-1273 vaccine. After the evaluation of humoral immunity to SARS-CoV-2, 14 seronegative patients were tested with two commercial IGRAs (SD Biosensor and Euroimmun). Out of 14 patients, one and three samples were positive by IGRAs with Euroimmun and SD Biosensor, respectively. The overall agreement between the two assays was 85.7% (κ = 0.444). In addition, multivariate linear regression analysis showed no statistically significant association between the IFN-γ concentration, and the independent variables analyzed (age, gender, years since transplant, total lymphocytes cells/mcl, CD3+ cells/mcl, CD3+ CD4+ cells/mcl, CD3+ CD8+ cells/mcl, CD19+ cells/mcl, CD3-CD16+CD56+ cells/mcl) (*p* > 0.01). In a vulnerable setting, assessing cellular immune response to complement the humoral response may be advantageous. Since the two commercial IGRAs showed a good agreement on negative samples, the three discordant samples highlight the need for further investigations.

## 1. Introduction

Since December 2019, the Severe Acute Respiratory Syndrome CoronaVirus-2 (SARS-CoV-2) has spread worldwide, causing coronavirus disease-2019 (COVID-19) [1]. Some comorbidities, such as cardiovascular disease, diabetes, and chronic kidney disease (CKD), have been shown to be predictors of mortality [2,3]. In addition, solid organ transplant recipients (SOTRs) have been at greater risk of severe disease and mortality compared to immunocompetent individuals. In particular, the impact of COVID-19 in the pre-vaccination period of the pandemic has been significant on kidney transplant recipients (KTRs), mainly in the early post-transplant period, with inpatient mortality rates ranging from 18% to 46% [4,5,6,7].

Starting in 2020, clinical trials have shown that mRNA vaccines are efficacious in preventing COVID-19 with no safety issues ascertained in immunocompetent individuals. SOTRs have been prioritized for vaccination despite their absence from the two large trials of Pfizer/BioNTech (BNT162b2) and Moderna (mRNA-1273) mRNA vaccines; therefore, safety and efficacy data are lacking in this population [8,9]. It is acknowledged that SOTRs, depending on the type of immunosuppressive regimen, evoke reduced immunogenicity towards other vaccines due to the repression of lymphocyte activation, interplay with antigen-presenting cells, and reduced B-cell memory responses [10,11,12,13,14]. Accordingly, during the pandemic, measuring the immune response of the SOTRs population to SARS-CoV-2 vaccines has been a major topic and has led to the investigation of both humoral and, to a minor extent, cellular responses after vaccination [15].

Vaccines, as well as natural infections, cause an early immune response [16,17] and stimulate long-lasting B- and T-cell memory responses in immunocompetent individuals [18,19]. Although the significance of the T-cell immune response in SARS-CoV-2 infection is not fully elucidated [20], evidence from animal models suggests that CD4+ and CD8+ T cells, as well as the production of interferon-gamma (IFN-), may play a key role in SARS-CoV-2 protection [21]. For this purpose, today, several tests to characterize the SARS-CoV-2-specific T cell response have been developed using different techniques. Notably, these include the enzyme-linked immune SPOT (ELISpot), flow cytometry, and enzyme-linked immunosorbent assay (ELISA) [22,23].

Cellular immune responses to SARS-CoV-2 in SOTRs have been described in two studies [24,25]. In particular, Candon et al. [24] showed how, after a reduction in immunosuppressive therapy, there were similar populations of SARS-CoV-2–reactive T cells in KTRs and non-transplant patients on hemodialysis with confirmed COVID-19. Moreover, Hartzell et al. [25] showed that KTRs with COVID-19, like non-transplant patients, had T lymphopenia, with a preference for CD8+ T lymphopenia, and that there was no emergence of exhausted, anergic, or senescent T cell populations. In addition, the authors were unable to discriminate between transplant patients with mild, moderate, or severe COVID-19 disease in terms of CD4+ or CD8+ T cell responses. Furthermore, according to the existing evidence on humoral responses to SARS-CoV-2, SOTRs may mount SARS-CoV-2–specific antibodies within 1–2 weeks of the onset of COVID-19 symptoms, and these antibodies can last for at least 2-months and possibly up to 4–6 months after infection, regardless of the severity of sickness [25,26,27,28]. Based on these data, SOTRs seem to induce humoral and cellular responses in a comparable scale to non-transplant patients. However, a body of evidence suggests a suboptimal humoral and cellular response, compared to immunocompetent individuals, after two doses and an additional booster dose of mRNA-based SARS-CoV-2 vaccine in SOTRs, including KTRs [15,29]. This is of concern among the transplant physician’s community, especially in the presence of viral variants with spike mutations, possibly associated with immune escape.

The aim of our study was to assess the SARS-CoV-2 directed T-cell response using two different interferon-gamma release assays (IGRAs) by ELISA in a cohort of KTRs with absent antibody production after the third dose of the mRNA-1273 vaccine. Moreover, we correlated the obtained IFN-γ results with independent variables obtained from the patients.

## 2. Results

All 14 samples that were tested for the two commercial IGRAs to detect T cell immune responses were classified as positive or negative based on the manufacturer’s cutoff values for IGRA Euroimmun and SD Biosensor.

Comparing the two IGRAs showed that one sample was positive by IGRA Euroimmun and three were positive by SD Biosensor.

In particular, the analysis showed eight double negative samples and one double positive. Three samples showed an invalid result with the Euroimmun assay due to a low mitogen value and a negative result with the SD Biosensor assay. Finally, two samples were positive by the SD Biosensor and negative by the Euroimmun assay (Table 1). However, when assimilating the invalid result to a negative one for the IGRA Euroimmun, the overall agreement between the two assays was 85.7% (12/14; κ = 0.444), the positive agreement 50%, and the negative agreement 91.7%.

Finally, we performed a multivariate linear regression analysis with all variables under investigation (age, gender, years since transplant, total lymphocyte cells, CD3+ cells/mcl, CD3+ CD4+ cells/mcl, CD3+ CD8+ cells/mcl, CD19+ cells/mcl, CD3-CD16+CD56+ cells/mcl). No statistically significant association was observed between the INF-γ concentration and the independent variables (*p* > 0.01).

## 3. Discussion

In our KTRs cohort, the rate of seroconversion after the third dose of the mRNA-1273 vaccine was lower (70.2%) than that of the general population [30]. This result is consistent with the data available in the literature, ranging from 40% to 70% [31,32,33,34,35,36,37,38,39,40,41,42,43,44,45]. KTRs are generally considered low responders to vaccines due to immunosuppressive therapy [46,47,48,49] and for this reason, the Italian National Institute of Health suggested specific COVID-19 surveillance for them, including both humoral and cellular immune response assessments [50].

Observations from case series studies in SOTRs show that some seronegative individuals develop at least a cellular response that could provide some protection against SARS-CoV-2 [35,42,51,52,53]. In fact, a robust T-cell response is typically one of the effects of COVID-19, and it seems to take an active role in terms of long-term immunological memory [54]. Actually, information on T-cell responses should be included to have a complete immunological examination and, in addition, should be part of a customized management approach in a population with lower post-vaccine antibody response.

On these premises, our study considered the evaluation of SARS-CoV-2-directed T cell responses in KTRs with an absence of antibody production after the third dose of the mRNA1273 vaccine. We used two commercial IGRAs, both ELISA-based, to evaluate the production of IFN-γ from T cells following vaccination. Our comparison analysis, given the small and not well distributed cohort of patients in terms of positive and negative samples, has a low statistical value and so few robust considerations can be drawn. However, the study highlighted how some differences between the two assays exist in relation to the validation rules (for example, in the definition of low mitogen, which determines an “invalid result”), the antigenic composition of the spike protein and the cutoff used. Although the Euroimmun manufacturer’s cutoff is 100 mIU/mL, some recent studies suggested the use of a lower cutoff based on their Receiver-Operating Characteristic (ROC) analysis [55,56]. Moreover, the SD Bionsensor assay may detect not only cell-mediated immune responses to SARS-CoV-2 (both to spike and nucleocapsid proteins ((N Protein)) but also its specific variants. Because in this study the vaccine immunogenicity of subjects was evaluated with no history of COVID-19 infection, possible differences regarding the antigenic composition of the assays should be of modest relevance. Interestingly, we detected one asymptomatic patient who was positive for the N protein. This outcome raises the question of whether this patient was immunized by direct contact/infection with the virus without presenting symptoms, or regardless of the vaccination process and the potential S protein immunization that was developed after the vaccination or not. Another interpretation of the positive response to the N protein may be attributed to the presence of cross-reactivity phenomenons with N proteins from other coronavirus members.

Furthermore, in our study, the evaluation of cellular responses through the IGRA showed a low percentage among seronegative patients after the third dose, with only one and three positive patients out of the 14 total patients, considering Euorimmun or SD Biosensor IGRAs, respectively. Currently, little data are available about T cell responses to SARS-CoV-2 vaccines in KTRs. It was observed that, in a study of 39 KTRs, the antibody response to mRNA vaccination was weak, but >80–90 percent of KTRs mounted CD4+ and CD8+ T cell responses [52], which were accompanied by a widespread impairment in effector cytokine production, memory differentiation, and activation-related characteristics. Importantly, studies from individuals with moderate COVID-19 also demonstrate that an early activation of IFN-T cells plays an important role in viral clearance [52].

Moreover, after the third dose of BNT162b2 vaccination, Bertrand et al. examined a large cohort of 80 KTRs for both humoral and T-cell responses, finding that 61% of KTRs had anti–spike IgG antibodies, and 70% had a significant number of IFN-producing spike-reactive T cells [35]. Although we did not evaluate IGRAs in all KTRs of our cohort, these data are comparable with our results in seronegative patients.

To what extent cellular immunity is able to prevent severe infection or death from SARS-CoV-2 in the absence of detectable antibodies is yet to be established, and only the clinical follow-up of these patients will indicate an exhaustive response. This is especially important for elderly KTRs with immune senescence related to thymus involution and, therefore, alterations in the number and proportion of the lymphocyte populations, and for KTRs in the early post-transplant period, especially in those with an anti-thymocyte globulin induction therapy [57]. A recent large study showed that the risk of COVID-19-related deaths in KTRs was 78% higher than in hemodialysis patients, particularly during the first post-transplant year [58].

Another interesting finding in our cohort is represented by the three cases of invalid results by Euroimmun IGRA. A recent study showed that an indeterminate test result (no measurable IFN-γ in the mitogen tube) was associated with severely ill COVID-19 patients compared to the control patients [59]. This defect of IFN-γ production was unrelated to absolute lymphocyte counts [59]. Indeed, the SARS-CoV-2 infection seems to disrupt the ability of peripheral T cells to generate IFN-γ via direct or indirect mechanisms. In the case of our KTRs, however, with no previous SARS-CoV-2 infection, the negative and invalid results had to be traced exclusively to their particular immunologic status.

In another recent study by Coppock et al., [60] indeterminate results suggested T cell exhaustion if IGRA data were taken as a surrogate for T cell functions because of the low-mitogen response. A negative test, given the typical mitogenic response, may imply a less- or non-impaired T-cell activity. The mitogen response in IGRA has been suggested as a surrogate for T cell functions and as a disease severity marker in COVID-19 [61]. Although evidence has shown suboptimal humoral and cellular immune responses to COVID-19 vaccinations in immunocompromised individuals [62,63], there is a lack of information concerning the over-time evaluation of both T- and B-cell responses in KRTs. Correlates of protection (COP) against SARS-CoV-2 infection have yet to be identified, and we have only determined B- and T-cell parameters so far [64,65,66]. As a result, based on the global detailed B- and T-cell responses, patients can be classified as full responders (both humoral and cellular), not responders, or partial responders (with only a humoral or T-cell response), similar to what has currently been suggested for rheumatoid arthritis (RA) patients [67,68]. Patients with a “not responder” or “partial responder” profile could use anti-SARS-CoV-2 monoclonal antibodies for prophylaxis, defined immunogenic peptide epitopes [69], or a tailored approach to risk management. This indication is anticipated to be supported by unpublished results from the “prevent” research, which confirms significant efficacy and long-term prevention [70].

Finally, in a vulnerable setting with lower seroconversion rates than the general non-immunocompromised population, assessing the cellular immune response may be extremely advantageous and noteworthy. As such, while many viral variants of concerns (VOCs) can escape humoral immunity, vaccine-induced cellular responses show a strong cross-protection against VOCs. This finding lends support to the idea that cellular responses play a major role in disease regulation by supporting tissue damage control [71]. The capacity of KTRs to mount a protective antiviral response, despite immunosuppression, would be critical during SARSCoV-2 infection. Despite complete immunization, partial and/or non-responders are very likely to be at higher risk for COVID-19, underscoring the significance of a targeted strategy.

However, our study presents some limitations. First, we restricted our investigation to KRTs and did not include healthy controls for a reference, as we were aware that there are a huge amount of data on the general population’s response to the SARS-CoV-2 infection. Secondly, we studied the T cell immunity only in seronegative patients rather than in the entire KTRs population. Lastly, this is a single center study with a low number of recruited patients and only a small number of positive IGRA results, which may limit the impact of the comparison investigation.

In conclusion, the two IGRAs showed a T cell response only in a poor rate of KTR subjects without any significant association between the INF-γ concentrations and demographic, anamnestic, and cytometic variables evaluated. However, future studies are needed to further assess their diagnostic performances, especially for screening applications in vulnerable populations at risk.

## 4. Methods and Materials

### 4.1. Patients

An observational study was conducted on a cohort of 57 KTRs who were actively followed up in the outpatient clinic of the Nephrology Division, Santa Maria Annunziata Hospital, Florence, Italy, with no history of COVID-19 infection. They received a 3rd mRNA-1273 vaccine dose six months after the 2nd dose, as suggested by local policy.

After the evaluation of humoral immunity to COVID-19, 3–4 weeks after the 3rd dose, 17 KTRs patients who tested negative for anti-S1 IgG antibodies underwent an evaluation for lymphocyte subpopulations. Fourteen out of the seventeen seronegative patients were subsequently tested with two commercial INF γ-release assays to detect T-cell immune response. The main demographic, laboratory and clinical characteristics of the studied population are reported in Table 2.

### 4.2. Anti-SARS-CoV-2 Antibody Detection

Anti-SARS-CoV-2 antibodies, directed to spike protein, were tested using the EliA SARS-CoV-2-Sp1 IgG (Thermo Fisher, Uppsala, Sweden). It is a fluoroenzyme-immunoassay (FEIA) for the quantitative detection of IgG antibodies in serums toward the SARS-CoV-2 spike 1 protein on the Phadia 250 instrument.

The EliA SARS-CoV-2-Sp1 IgG well is coated with a recombinant SARS-CoV-2 S1 protein. The cut-off value is 28 BAU/mL, as declared by the manufacturer.

### 4.3. Peripheral Blood Immunophenotyping

Immunophenotyping was performed using a BD Multitest 6-color TBNK reagent on the multicolor flow cytometer FACS CANTO II (BD Biosciences, San Jose, CA, USA).

### 4.4. First IGRA Assay (Euroimmun)

The Quant-T-Cell SARS-CoV-2 is a commercially available blood test system to determine quantitatively the IFN-γ released by SARS-CoV-2-specific T Cells (Euroimmun, Lübeck, Germany) [72,73].

Five hundred microliters of whole blood (in a lithium heparin tube) were transferred and gently mixed into one of a set of stimulation tubes: (1) CoV-2 IGRA TUBE, a stimulation tube coated with the S1 domain of the SARS-CoV-2 spike protein; (2) CoV-2 IGRA STIM, coated with mitogen as an unspecific control; and (3) CoV-2 IGRA BLANK, with no activating components for the immune system to determinate the patient background.

After 24 h of stimulation at 37 °C, all tubes were centrifuged at 12,000× *g* for ten minutes. The IFN-γ was measured in the supernatant plasma of all three aliquots using the Quant-T-Cell ELISA plate according to the manufacturer’s instructions (Euroimmun, Lübeck, Germany).

The microplate wells were coated with monoclonal anti-IFN-γ antibodies. In the first step calibrators, the controls, and plasma samples were diluted in sample buffer and added to the microplate to bind IFN-γ. In the second step, a biotin-labeled anti-IFN-γ antibody was added to detect the concentration of the IFN-γ antigen in the sample, expressed as milli-international units per milliliter (mIU/mL). The limit of the BLANK was 8.76 mIU/mL.

The results above the concentration of the highest calibrator were reported as >5 * calibrator 6 mIU/mL, as no further dilutions were performed.

A patient result was considered valid after checking for the validity of BLANK and STIM conditions. The IFN-γ concentration in the retrieved plasma of the BLANK had to be below the validity threshold of BLANK ≤ 400 mIU/mL.

Therefore, the IFN-γ concentrations of plasmas obtained from the tubes of the stimulation control STIM had to be above the validity threshold of STIM ≥ 400 mIU/mL, after BLANK subtraction.

After checking for the validity of BLANK and STIM conditions, the SARS-CoV-2-specific TUBE value condition could be evaluated. BLANK represents the individual IFN-γ background. Therefore, this value was subtracted from the IFN-γ concentration of the plasma retrieved from the other stimulation conditions. Values > 200 mIU/mL (after BLANK subtraction) were considered positive, values between 100 and 200 mIU/mL borderline and values < 100 mIU/mL negative.

However, invalid measurements can occur due to specific conditions e.g., immunosuppressive disease or immunosuppressive therapy that can lead to a reduced number and functional capacity of lymphocytes. On the other hand, other conditions may lead to the spontaneous release of IFN-γ-inducing and increased IFN-γ levels in the BLANK tube. In these cases, another test in 2–4 weeks is recommended.

### 4.5. Second IGRA Assay (SD Biosensor)

For all patients, 1 mL of whole blood was collected in 5 separate tubes (Covi-FERON Tubes, SD Biosensor, Inc., Gyeonggi-do, Korea): (1) nil tube, used to adjust for background noise and IFN-γ as a negative control; (2) original SP antigen tube, used to evaluate IFN-γ response to specific SARS-CoV-2 spike proteins (SP) from the 20I/501Y.V1 variant; (3) variant SP antigen tube, used to evaluate IFN-γ response to specific SARS-CoV-2 spike protein (SP) from 20H/501.V2 and 20J/501Y.V3 variants; (4) NP antigen tube, used to evaluate IFN-γ response to specific SARS-CoV-2 nucleocapsid proteins (NP); (5) mitogen tube, used as a positive control to check the patient’s immune status.

Following an incubation period from 16 to 24 h at 37 °C, the tubes were centrifuged, the plasma was collected, and the amount of IFN-γ (IU/mL) was measured by an enzyme-linked immunosorbent assay (STANDARD E Covi-FERON ELISA, SD Biosensor, Inc., Republic of Korea) according to the manufacturer’s instructions. IFN-γ concentrations were expressed as international units per milliliter (IU/mL). The accuracy of the test, which depends on the generation of an accurate standard curve, was examined before the interpretation of the samples’ outcome. Test results were interpreted qualitatively as reactive, non-reactive, or indeterminate according to the manufacturer’s instructions. Results of the STANDARD E Covi-FERON ELISA were determined according to the released cut-off and criteria. A “Reactive” result indicates that the sample contains an effector T-cell-mediated immune response to SARS-CoV-2, while a “Non-Reactive” result indicates that no effector T-cell-mediated response to SARS-CoV-2 was identified.

The results were reported as reactive when the IFN-γ value of the SP antigen (either the original SP or variant SP) tube or NP antigen tube minus that of the nil tube (nil value) was at least 0.25 IU/mL and 25% of the nil value. Moreover, the results were reported as indeterminate if the nil value was higher than 8.0 IU/mL or the value of the mitogen tube (mitogen value) minus the nil value was lower than 0.50 IU/mL and the IFN-γ value of the SP antigen (either the original SP or variant SP) tube and NP antigen tube minus the nil value were lower than 0.25 IU/mL or at least 0.25 IU/mL and lower than 25% of the nil value.

Table 3 illustrates the main features of each IGRA related to the manufacturer’s instructions.

### 4.6. Statistical Analysis

The agreement among methods was calculated by computing for each sample and the overall agreement score was set at the manufacturers’ cut-off value. In addition, we performed a multivariate linear regression to assess the relationship between the IFN-γ concentration and the variables under study (sex, age, total lymphocytes cells/mcL CD3+ cells/mcL, CD3+ CD4+ cells/mcL, CD3+ CD8+ cells/mcL, CD19+ cells/ mcL, CD3-CD16+CD56+ cells/ mcL, years since transplantation); an alpha level of 0.05 was considered to be statistically significant.

## Figures and Tables

**Table 1 ijms-23-12333-t001:** Comparison of results (raw data, quantitative and qualitative) of the two IGRAs.

Sample ID	Patient Initials	Test	BLANK	S Tube	Stim Tube	S Variant Tube	NC Tube	QuantitativeResult (mIU/mL)	QualitativeResult
1	SO	EUROIMMUN	19.67	82.76	>max	-	-	63	Negative
		SD BIOSENSOR	0.111	0.126	>10	0.071	0.008	-	Non-reactive
2	LL	EUROIMMUN	25	23.74	>max	-	-	0	Negative
		SD BIOSENSOR	0.099	0.088	>10	0.033	−0.011	-	Non-reactive
3	BLT	EUROIMMUN	14.27	72.04	689.32	-	-	57.8	Negative
		SD BIOSENSOR	0.069	0.418	>10	0.371	0.021	-	Reactive
4	SM	EUROIMMUN	26.49	61.23	>max	-	-	35	Negative
		SD BIOSENSOR	0.058	−0.005	>10	0.018	X	-	Non-reactive
5	LT	EUROIMMUN	44.04	43.19	94.8	-	-	Invalid	Invalid
		SD BIOSENSOR	0.232	−0.139	1.267	−0.111	−0.146	-	Non-reactive
6	VD	EUROIMMUN	22.47	102.76	1662.49	-	-	80	Negative
		SD BIOSENSOR	0.109	0.091	>10	−0.025	−0.04	-	Non-reactive
7	IC	EUROIMMUN	34.13	42.72	145.67	-	-	Invalid	Invalid
		SD BIOSENSOR	0.298	−0.166	1.039	−0.16	−0.222	-	Non-reactive
8	CF	EUROIMMUN	73.42	99.27	>max	-	-	26	Negative
		SD BIOSENSOR	0.173	−0.033	>10	−0.026	−0.048	-	Non-reactive
9	SG	EUROIMMUN	51.2	142.15	>max	-	-	91	Negative
		SD BIOSENSOR	0.115	0.142	>10	0.065	0.023	-	Non-reactive
10	MG	EUROIMMUN	22.34	26.72	>max	-	-	4	Negative
		SD BIOSENSOR	0.055	−0.003	>10	0.011	0.132	-	Non-reactive
11	TM	EUROIMMUN	3.99	26.93	>max	-	-	22.9	Negative
		SD BIOSENSOR	0.107	0.465	>10	0.029	−0.035	-	Reactive
12	PO	EUROIMMUN	13.07	56.52	>max	-	-	43	Negative
		SD BIOSENSOR	0.105	−0.029	>10	−0.036	−0.053	-	Non-reactive
13	PM	EUROIMMUN	<min	3.99	104.82	-	-	Invalid	Invalid
		SD BIOSENSOR	0.077	−0.022	>10	−0.022	−0.024	-	Non-reactive
14	MB	EUROIMMUN	36.53	351.76	>max	-	-	315	Positive
		SD BIOSENSOR	0.138	0.302	>10	0.125	0.263	-	Reactive

Euroimmun data are expressed in mIU/mL adjusted to international reference material (NIBSC, 82/587). S, S VARIANT and NC tubes values are blank subtracted. S: S1 domain of the spike protein for Euroimmun and spike derived from SARS-CoV-2 and 20I/501Y.V1 variant (originator) for SD Biosensor; STIM: stimulation (mytogen); S VARIANT: spike protein antigen derived from 20H/501.V2 and 20J/501Y.V3 variants; NC: Nucleocapsid.

**Table 2 ijms-23-12333-t002:** Demographic, clinical and laboratory characteristics of 14 kidney transplant recipients.

N.		Age	Gender	Total Lymphocytes Cells/mcL	CD3+ Cells/mcL (%)	CD3+ CD4+ Cells/mcL (%)	CD3+ CD8+ Cells/mcL (%)	CD19+ Cells/mcL (%)	CD3-CD16+CD56+ Cells/mcL (%)	Transplant (YEARS)	Immunosoppressive Therapy	Comorbidity
1	SO	54	M	2249	1214 (53.9%)	608 (27%)	574 (25.5%)	92 (4.1%)	943 (42%)	1	TAC/MMF/CCS	hypertension, diabetes
2	LL	31	M	1779	1496 (83%)	521 (29%)	856 (47%)	4 (0.2%)	279 (15%)	14	TAC/CCS	MGUS, hypertension
3	BLT	43	F	1573	1079 (68%)	538 (34%)	530 (33%)	205 (13%)	289 (18%)	2	TAC/MMF/CCS	
4	SM	54	F	2720	2054 (74.8%)	1098 (40%)	893 (32%)	308 (11%)	358 (13%)	1	TAC/MMF/CCS	hypertension, diabetes, cardiovascular disease
5	LT	67	F	1353	1235 (91%)	908 (67%)	313 (23%)	28 (2.1%)	93 (7%)	11	TAC/MMF/CCS	
6	VD	78	M	1381	872 (62%)	692 (49%)	170 (12%)	173 (12%)	336 (24%)	16	TAC/CCS	hypertension, diabetes, COVID-19 history
7	IC	59	M	2007	1891 (94%)	706 (35%)	1101 (55%)	74 (3.7%)	42 (2.1%)	10	TAC/CCS	hypertension, diabetes
8	CF	48	M	1257	1041 (82%)	604 (48%)	347 (27%)	55 (4.4%)	161 (13%)	1	TAC/MMF/CCS	hypertension, cardiovascular disease
9	SG	64	M	2188	1667 (76%)	538 (24%)	1047 (47%)	286 (13%)	235 (11%)	1	TAC/MMF/CCS	HIV +
10	MG	48	F	1233	1001 (81%)	660 (53%)	300 (24%)	23 (1.8%)	209 (17%)	3	TAC/MMF/CCS	MGUS, hypertension, diabetes, cardiovascular disease
11	TM	47	F	1450	1115 (76%)	681 (47%)	406 (28%)	150 (10%)	185 (12%)	4	TAC/MMF/CCS	Hypertension, autoimmune thyroiditis
12	PO	72	M	1853	1339 (72%)	260 (14%)	911 (49%)	66 (3.6%)	448 (24%)	2	TAC/m-TORi/CCS	hypertension, diabetes
13	PM	66	M	2446	2198 (89%)	481 (19%)	1581 (64%)	110 (4%)	138 (5%)	4	TAC/MMF/CCS	BPCO, hypertension
14	MB	56	M	2652	1895 871%)	1304 (49%)	552 (21%)	179 (7%)	578 (22%)	2	TAC/MMF/CCS	MGUS, hypertension, diabetes

tacrolimus (TAC)/mycophenolate mofetil (MMF)/corticosteroids (CCS)/inhibitors of Mammalian Target of Rapamycin (mTORi).

**Table 3 ijms-23-12333-t003:** Characteristics of IGRAs.

Manufacturer	Kit Assay	Method	Antigen (Stimulation Tube)	Coating Plate	Assay Time	Calibrators No.	Dynamic Range	Positivity Criteria	Validity Criteria
Euroimmun	*Quan-T-Cell ELISA*	ELISA	S1 domain	monoclonal anti-interferon-γ antibody	3 h 20′	6	0.5–2000 mIU/mL *	negative: <100 mIU/mL borderline: 100–200 mIU/mL positive: >200 mIU/mL	BLANK ≤ 400 mIU/mL STIM minus BLANK ≥ 400 mIU/mL
SD Bionsensor	*STANDARD E Covi-FERON ELISA*	ELISA	spike protein antigen derived from SARS-CoV-2 and 20I/501Y.V1 variant (original SP antigen tube), spike protein antigen derived from 20H/501.V2 and 20J/501Y.V3 variants (variant SP antigen tube), nucleocapsid protein antigen (NP Antigen tube)	monoclonal anti-interferon-γ antibody	Approx. 1 h 40′	4	0.125~10 IU/mL	Reactive: ≥0.250 IU/mL and ≥25% of Nil value	Nil ≤ 8.0 IU/mL Mitogen ≥ 0.5 IU/mL

* Dynamic range could vary between batch numbers.

## Data Availability

The data that support the findings of this study are available under reasonable request to the corresponding author MI.

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
