# Peer review of "Assessing T-Cell Immunity in Kidney Transplant Recipients with Absent Antibody Production after a 3rd Dose of the mRNA-1273 Vaccine"

_ijms, 2022, doi:10.3390/ijms232012333_

Round 1

Reviewer 1 Report

1)      LINE-21: Terminology needs to be changed, author used the terms Immune Surveillance, this could be confusing because Immune surveillance is also a term biologically used to describe host protective patho-physiological mechanisms .

2)      LINE-64: Check the abbreviation Interferon gamma

3)      LINE- 336 Check the word “tomoarticipate”

Author Response

1)      LINE-21: Terminology needs to be changed, author used the terms Immune Surveillance, this could be confusing because Immune surveillance is also a term biologically used to describe host protective patho-physiological mechanisms .

According to the suggestion of the reviewer, we changed "Immune surveillnze" into "Immune monitoring"

2)      LINE-64: Check the abbreviation Interferon gamma

We corrected it in the text

3)      LINE- 336 Check the word “tomoarticipate”

WE corrected it in the text

Reviewer 2 Report

In the current study, the authors investigated the vulnerable population of kidney transplants (KTRs) after 3rd dose of the Moderna vaccine in Italy and also they used two different interferon-gamma [IFN] release assay (IGRA). This is an interesting study but some parts of the text need some correction. However, I think after consideration of these comments, this work will be suitable for the current Journal. 

Comments to authors: 

** I recommend preparing a graphical abstract to improve the understanding of your study. 

** All changes in the manuscript should be identified in red font and yellow highlight. 

- Line 36: All keywords should be provided according to MeSH terms. 

- I suggest using this article to improve your introduction section: 

DOI: 10.1016/j.genrep.2021.101417. 

- Line 66: "developed using different technique" Please check the grammar. 

- Line 81: Please delete the point between "severity. [25]"

- Line 88: In this scenario "scenario" is not a reliable word for scientific writing. Please change or delete it. 

- Line 102: negative bu Euroimmun. "bu" change to "by"

- Line 142: All of the abbreviations should be complete at first use in the text and then use the abbreviation. Such as ROC 

- Lines 221-224: The conclusion must answer the aims set out in the introduction and also must be justified and logical. Should be modified. 

- Line 225: Please use reliable reference(s) for confirming your method section. 

- Line 259: Use one space between a number and its unit. Check it in all the text again. 

Author Response

- Line 36: All keywords should be provided according to MeSH terms.

We checked the keywords and changed them according to MeSH terms.

- I suggest using this article to improve your introduction section:

DOI: 10.1016/j.genrep.2021.101417.

We thank the reviewer; we inserted the mentioned article in the introduction.

- Line 66: "developed using different technique" Please check the grammar.

We corrected it in the text

- Line 81: Please delete the point between "severity. [25]"

We corrected it in the text

- Line 88: In this scenario "scenario" is not a reliable word for scientific writing. Please change or delete it.

We delete it

- Line 102: negative bu Euroimmun. "bu" change to "by"

We corrected it

- Line 142: All of the abbreviations should be complete at first use in the text and then use the abbreviation. Such as ROC

We corrected in the text

- Lines 221-224: The conclusion must answer the aims set out in the introduction and also must be justified and logical. Should be modified.

We modified the text accordingly

- Line 225: Please use reliable reference(s) for confirming your method section.

We added two references (72, 73) in the methods section

- Line 259: Use one space between a number and its unit. Check it in all the text again.

We checked the text again.